# A New Perspective on the Evaluation of Urbanization Sustainability: Urban Health Examination

**Hang Yan * and Zhijiang Liu**

School of Civil Engineering and Architecture, Wuhan University of Technology, Wuhan 430070, China;
334152@whut.edu.cn
* Correspondence: 13141@whut.edu.cn; Tel.: +86-15215195103

**Abstract:** In recent decades, urbanization around the world has become an unavoidable trend. However, rapid urbanization has brought about a number of "urban diseases" which have considerable influence on sustainable urbanization. In order to diagnose urban problems, this study introduces a new perspective for the evaluation of urbanization sustainability named "urban health examination (UHE)" based on the urban lifeform theory which treats a city as a human body system. Then, an evaluation index system of UHE incorporating eight dimensions is constructed by referring to the existing authoritative indicator systems globally. Furthermore, a deviation maximization method and obstacle analysis method are integrated to evaluate urban development level and diagnose the urban diseases. Finally, in order to verify the feasibility of the UHE methodological framework, an empirical study was conducted in Wuhan, Central China. The results show that (1) the main urban diseases suffered by Wuhan City in 2010–2019 include traffic congestion, waterlogging, unsafe production, insufficient technological power, and excessive urban development; (2) the evolution mechanism of urban diseases in Wuhan has been explored. This study proposes a methodological framework of UHE which can successfully diagnose urban diseases, so that local urban managers adopt tailored strategies to prevent urban diseases and further achieve sustainable urban development goals.

**Keywords:** urban health examination; urban lifeform theory; deviation maximization; obstacle analysis; Wuhan

## 1. Introduction

In the past decades, urbanization has become an unavoidable worldwide trend. According to the World Cities Report 2022 released by UN-Habitat, the percentage of people living in city areas worldwide increased from 30% to 56% between 1950 and 2021. It is projected that the urban population will exceed 60% of the global population, and the global city area will reach to $1.7 \times 10^6 \text{km}^2$ by 2050 [1]. Along with accelerating economic growth and social progress, rapid urbanization also brings about a number of challenges, including serious environmental damage [2–4], health problems [5,6], and income inequality [7–9]. These problems are also known as "urban diseases" which seriously hinder the quality of urban development. Therefore, it is critical to identify "urban diseases" and prescribe a "medicine" to address these "urban diseases".

The Chinese government has taken a number of initiatives to confront the growing "urban diseases" that accompany urban development. From 2016–2020, China began the first phase of a national healthy city pilot plan, which emphasizes the integration of urban public health issues into the core framework of urban planning [10]. Several scholars have reviewed critical social and environmental factors for healthy city construction and identified major stakeholders to provide schemes for implementing a new people-centered urbanization strategy [11]. However, in order to conduct a comprehensive scientific evaluation of cities, the Chinese government has proposed a more ambitious "urban health examination" strategy to achieve high-quality urban development. Similarly to human

physical examination, the primary purpose of the UHE is to diagnose "urban diseases" associated with urban development. Beijing took the lead in conducting UHE in 2018 as a pilot city. Beijing comprehensively monitored and evaluated the effects of the overall urban construction planning, and scientifically and systematically explored the problems existing in the urban construction process. The policy requirements of Beijing Master City Plan (2016–2035) are designed in accordance with these problems. As the first pilot city in China, Beijing has successfully achieved the scheduled goals of the UHE, and provided a model for the subsequent work of UHE. With the success of the UHE in Beijing, the number of pilot cities was increased from 11 to 36 in the following two years. Until 2021, the scale of the pilot cities was expanded to 59 to further assess the status of urban construction (as shown Figure 1).

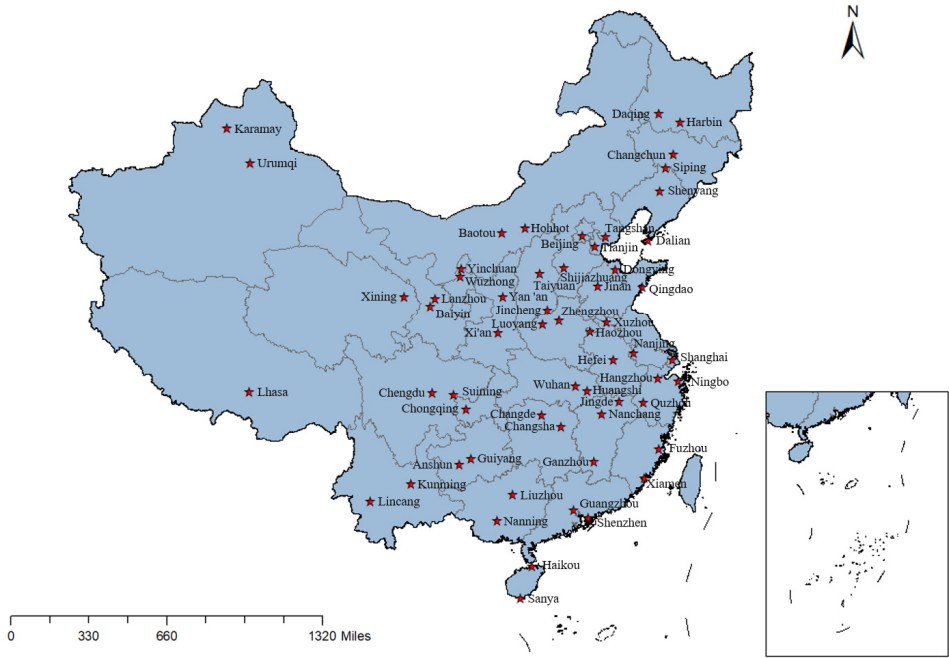

**Figure 1.** Pilot cities of urban health examination in 2021.

Numerous studies have been carried out to evaluate urban performance from a variety of perspectives, such as the sustainability [12–15], urban resilience [16], low-carbon development quality [17], urban support resources level and environmental pressure level [18], urban transport sustainability [19], urban livable environment evaluation [20], urban comprehensive bearing capacity [21], infrastructure carrying capacity [22], urban ecological carrying capacity [23], and urban innovation ability [24,25]. Additionally, a number of researches focus on the residents' health in the consideration of urban planning and emphasized the integration of healthy residents' community construction with town planning [26,27]. For example, Lan et al. [28] discussed the relevance of healthy urban planning to public health in four dimensions: land use, spatial form, transportation mode, and green open space. Furthermore, some scholars also concentrated on identifying the barriers in the urban development process. For example, Wang et al. [29] built an obstacle model to identify the primary hurdles affecting the development of an urban ecosystem; Zhang et al. [30] examined the obstacle factors affecting water security in Beijing using an obstacle degree model. Chen [31] proposed an innovative method for identifying urban diseases by integrating the subjective and objective methods. In conclusion, these studies have accumulated a significant amount of expertise in the evaluation of urban growth and have provided a solid framework for our investigation. However, few studies have established a mechanism for diagnosing "urban diseases" and assessed urban performance from a medical standpoint which treats a city the same as a human body. Furthermore, few

studies have diagnosed urban problems from a systematic and comprehensive perspective or investigate the evolution laws of these urban problems. Therefore, this study establishes the methodology framework of the UHE by integrating deviation maximization method and obstacles analysis. Wuhan is utilized as a case to demonstrate the effectiveness of the UHE methodological framework and discover the evolution of urban diseases over a period of time. Additionally, some targeted measures are introduced for the reference of local government.

The rest of this research is organized as follows: Section 2 analyzes the common characteristics that are present in cities and human bodies, and develops the theoretical foundation of UHE; Section 3 introduces the framework of UHE by integrating the deviation maximization method and the obstacle analysis method; Section 4 verifies the validity of the UHE method using Wuhan city as an example, and then diagnoses the urban diseases that have existed in Wuhan city in the past 10 years and proposes a "prescription"; followed by the conclusion in Section 5.

## 2. Theoretical Basis of Urban Health Examination

A city has many characteristics in common with the human body, as both are complicated systems [32]. For instance, cities and the human body are both composed of physical elements, similarly are the cells that make up our biological systems, which are frequently organized into repeating functional modules [33]. Furthermore, both of them are open systems that interact continually with the outside environment through the interchange of matter and energy [34]. In cities, the physical and biological processes of converting resources into useful products and wastes is similar to the human body's metabolic processes or that of an ecosystem [35]. The digestive system, respiratory system, circulatory system, endocrine system, and other systems make up the majority of the human body. Analogously, a city comprises various subsystems that keep it running smoothly [36]. Thus, this research studies cities with reference to the concept of living organisms, such as Meta et al. [37] who introduced urban physiology to explain the interactions between urban systems.

Similar to the human body, the city can be seen as an organic life form which composes complex interdependent systems across both "soft" and "hard" aspects. Specifically, cities generally contain systems for housing, transportation, sanitation, facilities, land use, and commodities. These systems have similar functions to the body system, for example transportation facility system in cities is very similar to the blood system which pumps blood around our body. The interwoven connection of the above urban subsystem facilitates the interaction between urban inhabitant, government bureaus, and private-sector, and benefits each stakeholder by improving product circulation efficiency and public service delivery during the urban development process [38]. However, this concentration can also induce significant negative consequences, such as generating urban heat islands, air pollution, and inefficient public services. Therefore, this paper proposes the UHE methodological framework based on the theory of urban lifeforms with reference to the process of human physical examination to help city governors understand the development status of the concerned urban context and diagnose urban diseases.

The city will suffer from "diseases" if one or more subsystems is running in disorder. Physical examination is typically used to detect diseases of the human body before the condition worsens [39]. In a similar way, the concept of a "physical examination" is introduced to the urban development area which follows the process of "diagnose diseases—treat diseases—consolidation improvement", as depicted in Figure 2. The UHE attempts to investigate the status of urban planning, urban construction, and urban management in order to identify "urban diseases". The local government can then take a variety of measures to treat the diseases discovered while maintaining high quality urban development.

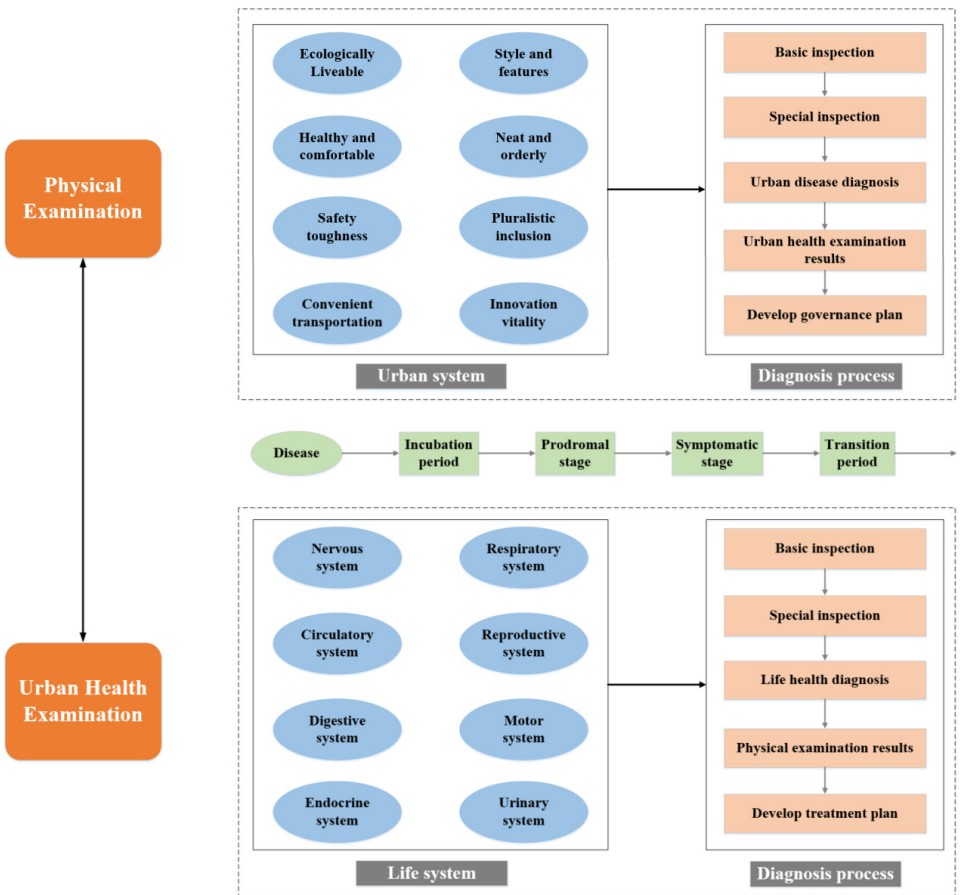

**Figure 2.** Methodological framework for urban health examination.

## 3. Methodology Framework

Based on the theoretical basis, this study establishes a methodology framework for UHE by integrating the deviation maximization method and obstacle factor analysis method. Firstly, an indicator system for UHE was constructed from eight dimensions. Second, a diagnostic model for "urban diseases" was proposed to diagnose the "urban diseases" of the target city, in which the deviation maximization method was applied to determine the weights of each indicator and obstacle factor analysis method was used to diagnose urban diseases. Finally, Wuhan City in central China was chosen as a case city to verify the effectiveness of the diagnostic model, and some policy suggestions were offered to Wuhan's administration to prevent the urban diseases. The technical framework for this investigation is shown in Figure 3.

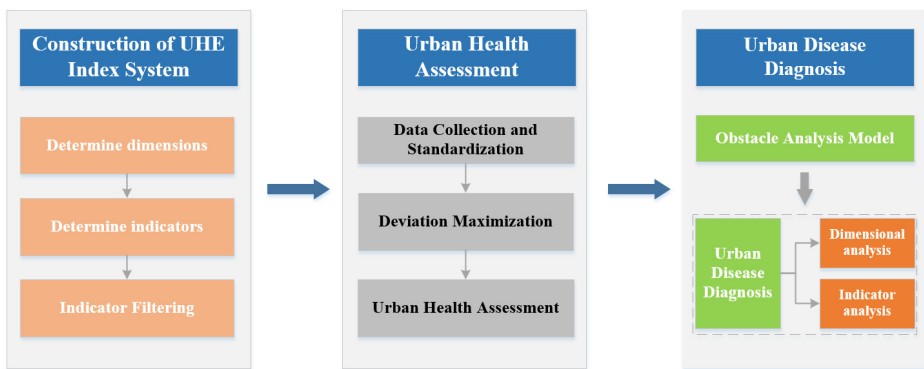

**Figure 3.** Methodology Framework.

### 3.1. The Construction of Indicator System for UHE

The situation of urban development is generated and perpetuated by a system of factors that span multiple levels and scales, including personal, social, environmental, and political factors [40]. Inspired by the bodily functioning systems (as illustrated in Figure 4), this study established a comprehensive indicator system for UHE from eight aspects: ecologically livable, health and comfortable, safety toughness, convenient transportation, style and features, neat and orderly, pluralistic inclusion, and innovation vitality. By combining the concepts of healthy cities and sustainable development, governments can raise the environmental concerns that are severely absent from sustainable urban development processes [41,42] and address the neglected or underestimated urban health focus [43,44]. Referring to existing authoritative indicator systems, such as the evaluation index system of the China Habitat Environment Award [45], the Sustainable Development Goals [46], Global Urban Indicators [47], Urban Indicators [48], this paper selects 48 indicators to optimize the existing UHE index system and constructs a more universal UHE index system (as shown in Table 1).

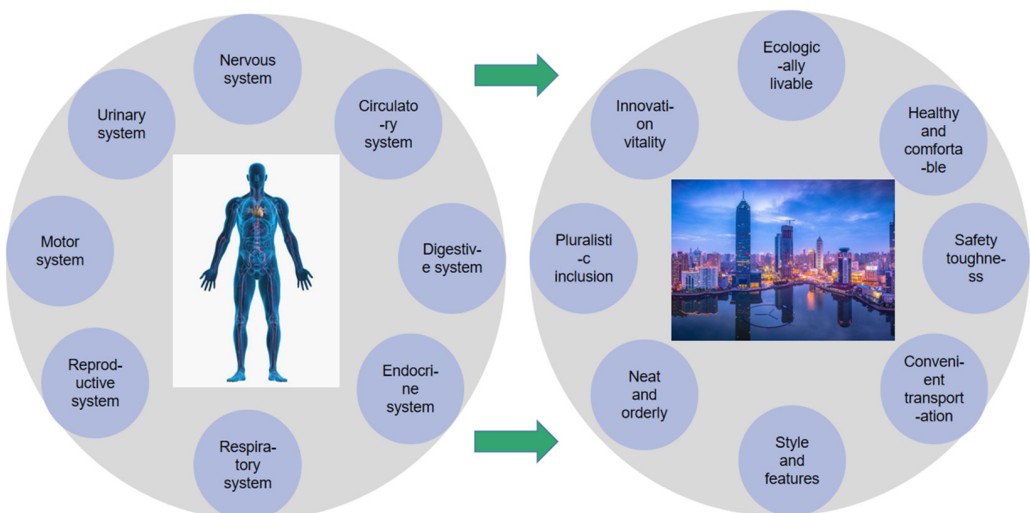

**Figure 4.** Human body system and urban health examination system [31].

**Table 1.** Urban health examination index system.

| Goal | Serial Number | Index | Indicator Type |
|------|---------------|-------|----------------|
| Ecologically livable (Z1) | C11 | Regional development intensity (%) | Interval type |
| | C12 | Urban population density (ten thousand people/km$^2$) | Intermediate type |
| | C13 | Green coverage rate in built-up area (%) | Positive type |
| | C14 | Centralized treatment rate of urban domestic sewage (%) | Positive type |
| | C15 | Harmless treatment rate of domestic garbage (%) | Positive type |
| | C16 | Days with good air quality (days) | Intermediate type |
| | C17 | Proportion of surface water reaching or better than class III water body (%) | Positive type |
| | C18 | Coverage rate of areas meeting regional environmental noise standards (%) | Intermediate type |
| | C19 | Comprehensive utilization rate of general industrial solid waste (%) | Positive type |

**Table 1.** *Cont.*

| Goal | Serial Number | Index | Indicator Type |
|---|---|---|---|
| Health and comfort (Z2) | C21 | Per capita collection of books in public libraries (volumes) | Positive type |
| | C22 | Number of doctors per 10,000 people (persons/10,000 people) | Positive type |
| | C23 | Per capita living area (square meters) | Positive type |
| | C24 | Sports area per capita (square meters) | Positive type |
| | C25 | Coverage rate of inclusive kindergartens (%) | Positive type |
| | C26 | Coverage rate of community service facilities for the elderly (%) | Positive type |
| | C27 | Number of beds in social welfare institutions for 10,000 people | Positive type |
| Safety toughness (Z3) | C31 | Number of sanitary beds per thousand population (pieces/thousand persons) | Positive type |
| | C32 | Death rate per 10,000 vehicles (%) | Negative type |
| | C33 | Density of waterlogging points in urban built-up areas (units/km$^2$) | Negative type |
| | C34 | Urban medical waste treatment capacity (%) | Positive type |
| | C35 | Safety mortality rate from production accidents per 100 million yuan GDP (%) | Negative type |
| Convenient transportation (Z4) | C41 | The proportion of investment in transportation construction in GDP (%) | Positive type |
| | C42 | Proportion of rail transit construction investment (%) | Positive type |
| | C43 | Traffic passenger volume of the whole society | Positive type |
| | C44 | Passenger turnover | Positive type |
| | C45 | Average vehicle speed during peak hours in built-up areas | Positive type |
| | C46 | Urban road network density | Interval type |
| | C47 | 800 m coverage rate of rail transit (%) | Positive type |
| | C48 | Per capita road area (square meters) | Positive type |
| Style and features (Z5) | C51 | Number of domestic and foreign tourists in the city (10,000 people) | Positive type |
| | C52 | Number of A-level scenic spots at the end of the year (units) | Positive type |
| | C53 | Number of cultural industry institutions per 10,000 people (units/10,000 people) | Positive type |
| Neat and orderly (Z6) | C61 | Centralized treatment rate of sewage treatment plant (%) | Positive type |
| | C62 | Actual cleaning area (ten thousand square meters) | Positive type |
| | C63 | Density of public toilets in built-up area (seats/km$^2$) | Positive type |
| | C64 | Drainage pipeline density in built-up area (m/km$^2$) | Positive type |
| Pluralistic inclusion (Z7) | C71 | House price to income ratio | Negative type |
| | C72 | The ratio of annual per capita disposable income to consumption expenditure of urban permanent residents | Positive type |
| | C73 | Urban employment rate (%) | Positive type |
| | C74 | Installation rate of road barrier-free facilities (%) | Positive type |
| | C75 | Participation rate of basic medical insurance for urban employees (%) | Positive type |

**Table 1.** *Cont.*

| Goal | Serial Number | Index | Indicator Type |
|---|---|---|---|
| Innovation vitality (Z8) | C81 | Number of high-tech enterprises per 10,000 people (units) | Positive type |
| | C82 | Number of newly-increased individual industrial and commercial households per 10,000 people | Positive type |
| | C83 | Number of listed companies per 10,000 people (units) | Positive type |
| | C84 | Number of postgraduate students per 10,000 people (a) | Positive type |
| | C85 | Percentage of permanent resident population | Positive type |
| | C86 | RD expenditure as a proportion of GDP (%) | Positive type |
| | C87 | The ratio of urban newly-added commercial housing to housing demand of the newly-increased population (%) | Positive type |

*3.2. The Overall Evaluation of Urban Health*

3.2.1. Standardization of Original Index Data

The indicators in the urban physical examination index system can be divided into four categories: positive type, negative type, intermediate type, and interval type. In order to facilitate the calculation, all indicators are consistently converted into positive indicators prior to examination. Definitions and conversions of each indicator type are as follows:

- Positive type

The positive indicator refers to the benefit-type indicator that is expected to be as large as possible. The other three types of indicators will be converted into positive indicators.

- Negative type

Negative indicators refer to indicators where a smaller value is better.

$$M = \max\{x_i\} - x_i \tag{1}$$

where $x_i$ is a set of negative indicator sequences and $\max\{x_i\}$ is the maximum value in this set.

- Intermediate type

Intermediate indicators are indicators whose optimal value is a specific value.

$$M = \max\{|x_i - x_{best}|\}, \widetilde{x_i} = 1 - \frac{|x_i - x_{best}|}{M}. \tag{2}$$

where $x_i$ is a sequence of intermediate type indicators with the best value of $x_{best}$.

- Interval type

If all values in an interval are optimal for the indicator, this type of indicator is called an interval type indicator.

$$M = \max\{a - \min\{x_i\}, \max\{x_i\} - b\}, \widetilde{x_i} = \begin{cases} 1 - \frac{a - x_i}{M}, x_i < a \\ 1, a \leq x_i \leq b \\ 1 - \frac{x_i - b}{M}, x_i > b \end{cases}. \tag{3}$$

where $x_i$ is a set of interval type indicator series with the optimal interval [a, b].

Additionally, the data must be converted in order to get the normalized judgment matrix because there are significant discrepancies between the magnitudes of the various

index values. Assume that the normalized matrix consists of n items to be evaluated and m orthogonalized indications to be evaluated as follows:

$$\begin{bmatrix} x_{11} & \cdots & x_{1m} \\ \vdots & \ddots & \vdots \\ x_{n1} & \cdots & x_{nm} \end{bmatrix}. \tag{4}$$

Then, the normalized matrix is Z, and each element in Z is represented as follows:

$$Z_{ij} = x_{ij} / \sqrt{\sum_{i=1}^{n} x_{ij}^2} \tag{5}$$

where $x_{ij}$ is the element in the normalized matrix.

3.2.2. Deviation Maximization Weighting Method

There are various weighting methods such as deviation maximization method, analytical hierarchy process (AHP), entropy method, genetic algorithms (GA), and neural networks (NN). This study adopts deviation maximization method to determine the weighting of indicators. The principle of deviation maximization method is to maximize the distance between different indicators. This approach has the advantage of amplifying indication variances and making it simpler to identify abnormal indicators. Therefore, the deviation maximization method has been widely used to determine weightings of indicators and multi-objective decisions. For instance, Yi et al. [49] used this method to rank the sustainable development levels of different cities. Li et al. [50] used it to calculate the indicator weights, which can amplify the overall differences between socioeconomic development and ecological environment. The detailed calculation procedure of deviation maximization is as follows:

The difference between different indicators can be expressed by the deviation, and the deviation of object i from object t is calculated by combining the results of the comprehensive evaluation of indicator j as:

$$d_{it} = \sum_{j=1}^{n} \omega_j |f_{ij} - f_{tj}| \tag{6}$$

where $f_{ij}$ is the evaluation value of object i under indicator j, $f_{tj}$ is the evaluation value of object t under indicator j and $\omega_j$ is the weight between evaluation indicators.

In the process of building the model, total division in all evaluations of objects is:

$$D = \sum_{i=1}^{m} \sum_{t=1}^{m} \sum_{j=1}^{n} \omega_j |f_{ij} - f_{tj}| \tag{7}$$

Based on the principle of deviation maximization, The weight vector $W = (\omega_1, \omega_2, \cdots \omega_n)^T$ could be obtained in such a way that the total deviation of all evaluation objects is maximum:

$$\max D = \sum_{i=1}^{m} \sum_{t=1}^{m} \sum_{j=1}^{n} \omega_j |f_{ij} - f_{tj}| \tag{8}$$

$$\text{s.t.} \sum_{j=1}^{n} \omega_j^2 = 1 \tag{9}$$

$$\omega_j > 0, j = 1, 2, \ldots, n \tag{10}$$

On this basis, a model based on the Lagrangian method is processed to determine the index weights by maximizing the deviation of the performance values. It is possible to obtain the following weights for each indicator:

$$\omega_j = \frac{\sum_{i=1}^{m} \sum_{t=1}^{m} |f_{ij} - f_{tj}|}{\sqrt{\sum_{j=1}^{n} \left( \sum_{i=1}^{m} \sum_{t=1}^{m} |f_{ij} - f_{tj}| \right)^2}} \tag{11}$$

Finally, the index weights $\omega_j$ need to be standardized so that they sum to 1, and the weights of each index of the final index system are obtained as:

$$\omega_j^* = \frac{\sum_{i=1}^{m} \sum_{t=1}^{m} \left| f_{ij} - f_{tj} \right|}{\sum_{j=1}^{n} \sum_{i=1}^{m} \sum_{t=1}^{m} \left| f_{ij} - f_{tj} \right|} \quad (i = 1, 2, \ldots, m) \tag{12}$$

The value of the evaluation index is proportional to its indicator weight, meaning that the larger the differential value, the more significant the indicator.

### 3.2.3. The Overall Performance of Urban Health

To make a comprehensive assessment of urban health, linear weighting method was adopted. The calculation formula is as follows:

$$S_{ij} = \sum_{j=1}^{n} \omega_j Z_{ij} \tag{13}$$

where $S_{ij}$ denotes the comprehensive health level of the city during, $\omega_j$ denotes the indicator weight, and $Z_{ij}$ denotes the standardized value of the indicator.

### *3.3. Obstacle Analysis Model*

The obstacle analysis model was first proposed by [51]. They defined the target as a set of expected behaviors and the obstacles as the undesirable behaviors that hinder target implementation. Usually, the identification of obstacles is mostly completed by calculating the level of contribution, deviation, and obstacles [27]. Some scholars have applied the obstacle analysis model to find the problems in the urbanization process. Chen and Zhang [52] evaluated the performance of the sustainable development of multiple cities and explored the main obstacle factors that restrict the sustainable development of the city. Wang et al. [26] adopted an obstacle analysis approach to identify the critical elements that prevent urban ecological health development. The TOPSIS approach and the obstacle model were merged by Wang et al. [53] to examine the primary barrier elements limiting the quality of low-carbon development in the city.

Therefore, this paper applied the obstacle analysis model to identify the main "urban diseases" which hinder the sustainable development of a city. With reference to the procedure of the physical examination, this research not only examines the degree of obstacles at the dimension level but also at the indication level. The unhealthy dimensions and indicators are all identified based on the degree of obstacles. Finally, tailored measures are proposed for these unhealthy indicators.

To evaluate the degree of the obstacle, the factor decomposition approach which includes factor contribution and index deviation were employed. The factor contribution degree indicates the influence of a single indicator on the total indicator, and this study used the weight $\omega_j$ of a single factor to express the factor contribution degree. The indicator deviation degree indicates the deviation between the actual indicator value and the optimal target value, and the difference between the normalized value of each indicator and 1 is used in this study, that is $1 - y_{ij}$. The obstacle degree $O_{ij}$ indicates that each indicator or guideline level factor affects the level of urban health of the city, and O is the obstacle degree of each dimension, which is calculated by the following formula:

$$O_{ij} = \frac{\left(1 - y_{ij}\right) * \omega_{ij}}{\sum_{j=1}^{n} \left(1 - y_{ij}\right) * \omega_j} \tag{14}$$

The obstacle score of subsystems:

$$O = \sum O_{ij} \tag{15}$$

**4. Case Study of Urban Health Examination**

*4.1. Study Area*

Wuhan City, which is situated in the middle and lower portions of the Yangtze River Plain, is the capital city of Hubei Province and a national regional core city of Central China. It is known as the "City of Hundred Lakes" and is situated at 113°41′–115°05′ east longitude and 29°58′–31°22′. Wuhan city, 25% of whose total area is water, has been a major transportation hub for urban development for more than 3500 years. By the end of 2022, the city had a population of 13.649 million and a regional GDP of CNY 1.89 trillion, which ranked the first among the Yangtze River midstream city group and ninth nationwide in the total economic output. Although Wuhan's economy has achieved rapid growth in recent years, it has also encountered common urban problems such as traffic congestion, environmental pollution, and soaring housing prices, especially since the outbreak of novel coronavirus (COVID-19) in early 2020, the city has suffered a series of severe challenges as the core city of the virus outbreak, exposing its shortcomings in various aspects such as medical resource allocation and urban security management. Therefore, it is necessary to conduct an in-depth analysis of the health performance of Wuhan city and comprehensively diagnose the underlying urban diseases so that the government can take tailored measures to effectively address the city's problems.

*4.2. Data Source*

In order to explore the health level of Wuhan City in the past decade, this study collected the data for each indicator (as listed in Table 1) of Wuhan City from 2010 to 2019 from statistical yearbooks and government bulletins, including Hubei Statistical Yearbook [54], Wuhan Statistical Yearbook [55], China City Statistical Yearbook [56], and Wuhan Environmental Status Bulletin [57].

*4.3. Results*

4.3.1. Index Weighting Value

Firstly, data standardization was performed to process the raw data collected according to Formulas (1)–(4). Then, the weight of each indicator can be acquired by using deviation maximization weighting methods according to Formulas (6)–(12). The weighting of the indicators' computation results is displayed in Table 2.

**Table 2.** Weight coefficients of each indicator.

| Indicator | Weight |
|---|---|
| Regional development intensity (%) | 0.0479 |
| Urban population density (ten thousand people/km$^2$) | 0.0316 |
| Green coverage rate in built-up area (%) | 0.0024 |
| Centralized treatment rate of urban domestic sewage (%) | 0.0022 |
| Harmless treatment rate of domestic garbage (%) | 0.0042 |
| Days with good air quality (days) | 0.0483 |
| Proportion of surface water reaching or better than class III water body (%) | 0.0059 |
| Coverage rate of areas meeting regional environmental noise standards (%) | 0.0470 |
| Comprehensive utilization rate of general industrial solid waste (%) | 0.0014 |
| Per capita collection of books in public libraries (volumes) | 0.0130 |
| Number of doctors per 10,000 people (persons/10,000 people) | 0.0065 |
| Per capita living area (square meters) | 0.0044 |
| Sports area per capita (square meters) | 0.0231 |
| Coverage rate of inclusive kindergartens (%) | 0.0256 |
| Coverage rate of community service facilities for the elderly (%) | 0.0244 |
| Number of beds in social welfare institutions for 10,000 people | 0.0267 |
| Number of sanitary beds per thousand population (pieces/thousand persons) | 0.0118 |
| Death rate per 10,000 vehicles (%) | 0.0453 |
| Density of waterlogging points in urban built-up areas (units/km$^2$) | 0.0519 |
| Urban medical waste treatment capacity (%) | 0.0196 |
| Safety mortality rate from production accidents per 100 million yuan GDP (%) | 0.0498 |
| The proportion of investment in transportation construction in GDP (%) | 0.0416 |

**Table 2.** *Cont.*

| Indicator | Weight |
|---|---|
| Proportion of rail transit construction investment (%) | 0.0278 |
| Traffic passenger volume of the whole society | 0.0074 |
| Passenger turnover | 0.0153 |
| Average vehicle speed during peak hours in built-up areas | 0.0224 |
| Urban road network density | 0.0560 |
| 800 m coverage rate of rail transit (%) | 0.0311 |
| Per capita road area (square meters) | 0.0093 |
| Number of domestic and foreign tourists in the city (10,000 people) | 0.0336 |
| Number of A-level scenic spots at the end of the year (units) | 0.0102 |
| Number of cultural industry institutions per 10,000 people (units/10,000 people) | 0.0106 |
| Centralized treatment rate of sewage treatment plant (%) | 0.0027 |
| Actual cleaning area (ten thousand square meters) | 0.0286 |
| Density of public toilets in built-up area (seats/km$^2$) | 0.0256 |
| Drainage pipeline density in built-up area (m/km$^2$) | 0.0021 |
| House price to income ratio | 0.0395 |
| The ratio of annual per capita disposable income to consumption expenditure of urban permanent residents | 0.0043 |
| Urban employment rate (%) | 0.0003 |
| Installation rate of road barrier-free facilities (%) | 0.0008 |
| Participation rate of basic medical insurance for urban employees (%) | 0.0029 |
| Number of high-tech enterprises per 10,000 people (units) | 0.0530 |
| Number of newly-increased individual industrial and commercial households per 10,000 people | 0.0136 |
| Number of listed companies per 10,000 people (units) | 0.0067 |
| Number of postgraduate students per 10,000 people (a) | 0.0092 |
| Percentage of permanent resident population | 0.0027 |
| RD expenditure as a proportion of GDP (%) | 0.0057 |
| The ratio of urban newly-added commercial housing to housing demand of the newly-increased population (%) | 0.0441 |

4.3.2. Urban Health Level of Wuhan from 2010 to 2019

The overall urban health level of Wuhan in the past ten years was obtained by adopting Formula (13) (as shown in Figure 5). It is evident that the urban health level of Wuhan city has significantly improved over the past ten years as the overall performance of Wuhan City has increased from 0.9431 to 1.4821 in past decades. It indicates that the Wuhan municipal government has taken effective measures to improve urban construction and urban management level in the past decade.

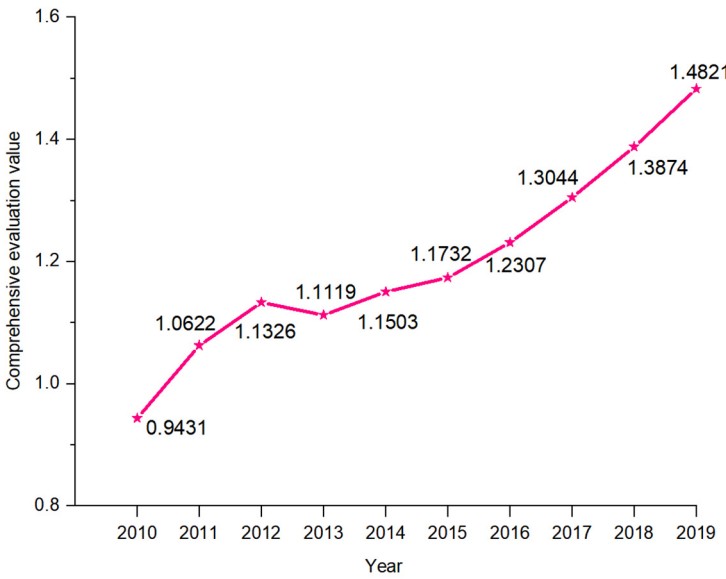

**Figure 5.** The overall urban health performance of Wuhan in the past 10 years.

### 4.3.3. Results of Obstacle Analysis

(1)   Obstacle analysis of eight dimensions

Figure 6 reflects the trend of the obstacle degree in 8 dimensions, and it can be seen that from 2010 to 2019, the obstacle dimensions with the unhealthy performance of Wuhan City are convenient transportation (Z4), ecologically livable (Z1), and safety toughness (Z3), with the sum of the factor contributions of these three dimensions exceeding 55% over the 10 year period. With the rapid development in the urban economy, Wuhan has encountered the similar traffic congestion problems as other large cities with a 10 million population. The traffic congestion has become a regular phenomenon during the morning and evening rush hours on weekdays, due to the over-density of population and unreasonable road design, which has led to the road congestion spreading from the central to the periphery of the city. The number of motor vehicles in Wuhan exceeded 4 million by the end of 2021, which not only presents a significant issue to urban traffic and security but also negatively impacts the quality of life for citizens due to the high amount of exhaust pollutants from cars. To achieve green development in Wuhan, the city administration must take necessary measures, including expanding public transit and supporting new energy vehicles. In addition, as the major manufacturing city in Central China, Wuhan contains 38 industrial categories, which include heavy polluting enterprises such as steel, automobile, petrochemicals, etc. Industrial wastewater and exhaust gas emissions pose a serious threat to the ecological environment of the city. Therefore, since Wuhan was chosen as the second batch of national pilot cities for low-carbon city construction in 2012, the city is continuously transforming traditional industries such as building materials, chemicals, iron, and steel with energy-saving and carbon-reducing technologies, and has achieved good performance in building a clean and low-carbon energy consumption structure. Urban security and production safety management are also significant issues in Wuhan; thus, the 2021 Work Report of the Wuhan Municipal People's Government has proposed to strengthen the construction of the national regional emergency rescue center, improve the flood control system, and implement special governance plans for safe production to improve the medical and health level of the city. The two dimensions of healthy and comfortable (Z2) and innovation vitality (Z8) have lower obstacle degree in Wuhan, which indicates that the Wuhan city have better performance in the construction of livable environment and enterprise innovative capability. The three dimensions of neat and orderly (Z6), style and features (Z5), and pluralistic inclusion (Z7) have the lowest obstacle degree, which means that the Wuhan city perform excellent in these three aspects.

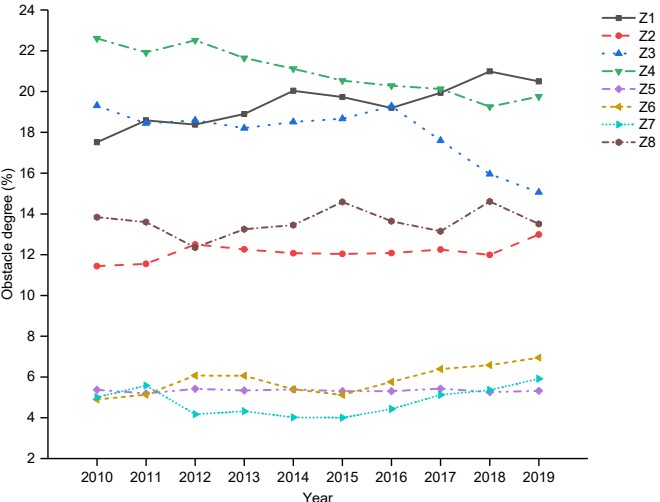

**Figure 6.** Dynamic change trend of obstacle degree of eight dimensions of urban health in Wuhan from 2010 to 2019.

(2)  UHE index-based obstacle analysis

By using Formula (14), the obstacle degree of each indicator during 2010 to 2019 can be obtained. Then, the annual average of obstacle degree for each indicator is listed in Table 3. The obstacle degree of each indicator represents the obstacle influence on the development of Wuhan city, and the deeper the color, the greater the degree of influence of the indicators. The top five indicators with obstacle degree above 5% are urban road network density, density of waterlogging points in urban built-up areas, safety mortality rate from production accidents per 100-million-yuan GDP, number of high-tech enterprises per 10,000 people, and regional development intensity. This result indicates that the main urban diseases suffered by Wuhan City in the past decade include traffic congestion, waterlogging, unsafe production, insufficient technological power, and excessive urban development. Wuhan's government should take corresponding measures to treat these cities diseases.

**Table 3.** Average obstacle degree of all indicators from 2010 to 2019.

| Indicator | Obstacle Degree |
|---|---|
| Urban road network density | 6.39% |
| Density of waterlogging points in urban built-up areas (units/km$^2$) | 5.96% |
| Safety mortality rate from production accidents per 100 million yuan GDP (%) | 5.63% |
| Number of high-tech enterprises per 10,000 people (units) | 5.38% |
| Regional development intensity (%) | 5.16% |
| The proportion of investment in transportation construction in GDP (%) | 4.83% |
| The ratio of urban newly-added commercial housing to housing demand of the newly-increased population (%) | 4.61% |
| House price to income ratio | 4.28% |
| Death rate per 10,000 vehicles (%) | 4.10% |
| Urban population density (ten thousand people/km$^2$) | 3.83% |
| Number of domestic and foreign tourists in the city (10,000 people) | 3.49% |
| Coverage rate of areas meeting regional environmental noise standards (%) | 3.23% |
| Days with good air quality (days) | 3.11% |
| 800 m coverage rate of rail transit (%) | 3.01% |
| Actual cleaning area (ten thousand square meters) | 2.88% |
| Proportion of rail transit construction investment (%) | 2.76% |
| Number of beds in social welfare institutions for 10,000 people | 2.63% |
| Coverage rate of inclusive kindergartens (%) | 2.53% |
| Coverage rate of community service facilities for the elderly (%) | 2.46% |
| Sports area per capita (square meters) | 2.34% |
| Average vehicle speed during peak hours in built-up areas | 2.25% |
| Urban medical waste treatment capacity (%) | 2.10% |
| Density of public toilets in built-up area (seats/km$^2$) | 1.89% |
| Passenger turnover | 1.49% |
| Number of newly-increased individual industrial and commercial households per 10,000 people | 1.36% |
| Number of sanitary beds per thousand population (pieces/thousand persons) | 1.25% |
| Number of cultural industry institutions per 10,000 people (units/10,000 people) | 1.14% |
| Per capita collection of books in public libraries (volumes) | 1.05% |
| Number of A-level scenic spots at the end of the year (units) | 0.98% |
| Per capita road area (square meters) | 0.93% |
| Number of postgraduate students per 10,000 people (a) | 0.88% |
| Traffic passenger volume of the whole society | 0.72% |
| Number of doctors per 10,000 people (persons/10,000 people) | 0.65% |
| Number of listed companies per 10,000 people (units) | 0.62% |
| Proportion of surface water reaching or better than class III water body (%) | 0.57% |
| RD expenditure as a proportion of GDP (%) | 0.55% |
| Harmless treatment rate of domestic garbage (%) | 0.43% |
| Per capita living area (square meters) | 0.42% |
| The ratio of annual per capita disposable income to consumption expenditure of urban permanent residents | 0.41% |
| Participation rate of basic medical insurance for urban employees (%) | 0.28% |
| Percentage of permanent resident population | 0.26% |
| Centralized treatment rate of sewage treatment plant (%) | 0.26% |
| Green coverage rate in built-up area (%) | 0.23% |
| Centralized treatment rate of urban domestic sewage (%) | 0.22% |
| Drainage pipeline density in built-up area (m/km$^2$) | 0.21% |
| Comprehensive utilization rate of general industrial solid waste (%) | 0.14% |
| Installation rate of road barrier-free facilities (%) | 0.08% |
| Urban employment rate (%) | 0.03% |

Then, an in-depth analysis on the changing trend of "urban diseases" in Wuhan is also conducted. The study period is divided into two parts, namely, from 2010 to 2014 and from 2015 to 2019 based on the Chinese government's practice of making a critical national economic and social development plan every five years. From Tables 4 and 5, the major obstacle indicators during the 12th Five-Year Plan period (2010–2014) include C46 (urban road network density), C33 (density of waterlogging points in urban built-up areas), C35 (safety mortality rate from production accidents per 100-million-yuan GDP), C81 (number of high-tech enterprises per 10,000 people), and C11 (regional development intensity). It indicates that Wuhan suffers the "urban diseases" of road traffic congestion, flood management, production safety, science and technology innovation, and development intensity. During the 13th Five-Year Plan period (2015–2019), the indicators with great obstacle degree are concentrated in C46 (urban road network density), C18 (coverage rate of areas meeting regional environmental noise standards), C32 (death rate per 10,000 vehicles), C87 (the ratio of urban newly-added commercial housing to housing demand of the newly increased population), and C16 (days with good air quality). The major "urban diseases" of Wuhan city in this period are traffic congestion, noise pollution, traffic safety, housing supply, and air pollution.

**Table 4.** Top five obstacles in 2010–2019.

| Index ranking | 2010 | | 2011 | | 2012 | | 2013 | | 2014 | |
|---|---|---|---|---|---|---|---|---|---|---|
| | Obstacle factor | Obstacle degree/% | Obstacle factor | Obstacle degree/% | Obstacle factor | Obstacle degree/% | Obstacle factor | Obstacle degree/% | Obstacle factor | Obstacle degree/% |
| 1 | C46 | 6.71 | C46 | 6.54 | C46 | 6.71 | C11 | 6.53 | C11 | 6.25 |
| 2 | C33 | 6.21 | C81 | 6.03 | C81 | 6.22 | C46 | 6.34 | C81 | 5.99 |
| 3 | C35 | 5.96 | C33 | 5.85 | C33 | 5.84 | C81 | 5.98 | C46 | 5.82 |
| 4 | C81 | 5.68 | C18 | 5.80 | C35 | 5.66 | C33 | 5.47 | C33 | 5.47 |
| 5 | C11 | 5.73 | C35 | 5.64 | C11 | 5.47 | C35 | 5.33 | C35 | 5.33 |

| Index ranking | 2015 | | 2016 | | 2017 | | 2018 | | 2019 | |
|---|---|---|---|---|---|---|---|---|---|---|
| | Obstacle factor | Obstacle degree/% | Obstacle factor | Obstacle degree/% | Obstacle factor | Obstacle degree/% | Obstacle factor | Obstacle degree/% | Obstacle factor | Obstacle degree/% |
| 1 | C32 | 6.43 | C33 | 6.20 | C18 | 5.96 | C18 | 6.79 | C87 | 5.83 |
| 2 | C11 | 6.07 | C32 | 5.51 | C46 | 5.17 | C87 | 6.36 | C16 | 5.71 |
| 3 | C81 | 5.73 | C46 | 5.41 | C32 | 5.12 | C16 | 5.24 | C18 | 5.39 |
| 4 | C46 | 5.70 | C81 | 5.37 | C81 | 5.01 | C46 | 4.69 | C71 | 4.95 |
| 5 | C87 | 5.12 | C18 | 5.22 | C16 | 5.01 | C71 | 4.48 | C46 | 4.49 |

**Table 5.** Frequency of major obstacles from 2010 to 2019.

| Year | Obstacle Factor | C46 | C33 | C35 | C81 | C11 | C18 | C31 | C32 | C87 | C16 | C71 |
|---|---|---|---|---|---|---|---|---|---|---|---|---|
| 2010–2019 | The number of occurrences | 10 | 6 | 5 | 8 | 4 | 5 | 1 | 3 | 3 | 3 | 2 |
| | Frequency of occurrence/% | 100 | 60 | 50 | 80 | 40 | 50 | 10 | 30 | 30 | 30 | 20 |
| 2010–2014 | The number of occurrences | 5 | 5 | 5 | 5 | 4 | 1 | 0 | 0 | 0 | 0 | 0 |
| | Frequency of occurrence/% | 100 | 100 | 100 | 100 | 80 | 20 | 0 | 0 | 0 | 0 | 0 |
| 2015–2019 | The number of occurrences | 5 | 1 | 0 | 3 | 0 | 4 | 1 | 3 | 3 | 3 | 2 |
| | Frequency of occurrence/% | 100 | 20 | 0 | 60 | 0 | 80 | 20 | 60 | 60 | 60 | 40 |

By comparing the "urban diseases" in these two periods, it can be found that the traffic congestion problem has always plagued Wuhan City in past decade. In recent years, the construction of the city's internal transportation infrastructure has made great progress. The length of railway has increased from 125 km to 410 km during the period of 2015 to 2020, and the mileage of urban roads has reached 2318 km by the end of 2020. However, with the rapid growth and sprawl of urban transportation infrastructure in Wuhan, some critical problems have also emerged. For example, there is an inherent deficiency in the urban road network. Although 15 river crossings have been built in Wuhan, it is still difficult to support the huge daily traffic flow in the city. Some new urban areas have insufficient road density, and the metro network does not yet cover some of the emerging key development areas. At the same time, as the core city in the central region, Wuhan

has a large number of universities and enterprises, and the city's population is growing rapidly. Despite the rapid construction of Wuhan's transportation infrastructure, it is still unable to meet the growth rate of the city's vehicles and population. Therefore, there is an urgent need for the Wuhan government to understand the shortages and deficiencies of transportation infrastructure from the perspective of the city's residents and to develop effective measures to properly address these problems.

Another impressive finding is that the urban diseases in Wuhan City have changed a lot during the 12th Five-Year Plan and 13th Five-Year Plan. As urbanization progressed, urban problems have shifted from improving infrastructure construction to improving the quality of life of residents. During the 12th Five-Year Plan period, the primary contradiction encountered by Wuhan was that the infrastructure construction could not keep up with its rapid development in urbanization. With the continuous improvement in infrastructure, the main contradiction of Wuhan City in the 13th Five-Year Plan is the increasing demand for people's better lives and the decrease in the urban living environment. This significant change is not only consistent with the objective rules of urban growth, but also with the macro policies of the Chinese government. In 2013, Chinese Central government first proposed a new type of urbanization policy based on the principle of "People-Centered". The focus of urbanization has transformed from "the economic growth and scale expansion" to "the increase in life quality and ecological civilization". Recently, Wuhan City is in the late period of urbanization, the infrastructure is relatively complete but the population is constantly flowing in, which caused environmental deterioration and decline in living quality. In summary, we can find that the evolution of urban diseases is closely correlated with the development stage of urbanization, and follows the rule of transforming from the improvement urban infrastructure to the improvement of urban human settlement environment.

## 5. Conclusions

Rapid urbanization not only improves people's living standards and quality, but also brings traffic congestion, environmental degradation, resource shortages and other "urban diseases". It is important to propose a scientific methodology to assess the health status of a city and diagnose the underlying problems of urban development, so that effective action can be taken to treat these diseases. In order to diagnose urban diseases, this study innovatively draws on the idea of physical examination to form an UHE methodological framework. An UHE index system with eight dimensions is constructed. The deviation maximization method is adopted to determine the weight of each index, and obstacle analysis method is used to identify "urban diseases". The applicability and effectiveness of the method are verified by a case study of Wuhan, Central China. The findings of this study demonstrate that the suggested method can assist decision-makers and urban planners in identifying urban diseases accurately and discovering that the evolution mechanism of urban diseases is consistent with urbanization process.

The main contributions of this study include two aspects. Theoretically, this study refers to medical theory and the process of human physical examination to construct the indicator system comprehensively. Furthermore, a method of data standardization is introduced for four types of indicator. Then, a scientific mechanism which combines the deviation maximization method and obstacle analysis method is proposed to diagnose urban problems. Practically, this study combines empirical analysis to demonstrate how the UHE method can be used to conduct a "physical examination" of cities. The results of the analysis show that the UHE framework can effectively diagnose urban diseases that exist in the process of urban development. Additionally, this mechanism can be used to investigate the evolution of urban diseases over a period of time and explain the impact of national macro-control policies on urban development. This helps local governments grasp the current status of urban development, accurately identify the shortcomings of cities, and implement targeted measures to improve the quality of the urban living environment.

However, there are also many aspects that need to be improved in the future study. Firstly, in order to diagnose more urban diseases, the indicator system of UHE can be further improved to cover more aspects of urban development; secondly, more cities can be further analyzed to prove the effectiveness and applicability of the proposed methodology.

**Author Contributions:** Conceptualization, H.Y.; Methodology, H.Y.; Validation, Z.L.; Data curation, Z.L.; Writing—original draft, H.Y.; Writing—review & editing, Z.L. All authors have read and agreed to the published version of the manuscript.

**Funding:** This research received no external funding.

**Institutional Review Board Statement:** Not applicable.

**Informed Consent Statement:** Not applicable.

**Data Availability Statement:** Data available in a publicly accessible repository. The data presented in this study are openly available in [Hubei Statistical Yearbook] at [http://tjj.hubei.gov.cn/tjsj/sjkscx/tjnj/qstjnj/index.shtml], reference number [54]; [Wuhan Statistical Yearbook] at [http://tjj.wuhan.gov.cn/tjfw/tjnj/], reference number [55]; [Chain City statistical Yearbook] at [https://data.cnki.net/v3/Trade/yearbook/single/N2022040095?zcode=Z024], reference number [56]; [Wuhan Environmental Status Bulletin] at [http://hbj.wuhan.gov.cn/fbjd_19/xxgkml/zwgk/hjjc/hjzkgb/], reference number [57]. (accessed on 26 May 2023).

**Conflicts of Interest:** The authors declare no conflict of interest.

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
