# Peer review of "A New Perspective on the Evaluation of Urbanization Sustainability: Urban Health Examination"

_sustainability, doi:10.3390/su15129338_

Round 1
Reviewer 1 Report
The aim of the work is the evaluation of urbanization sustainability named "urban health examination (UHE)" based on the urban lifeform theory which treat a city as a human body system.
Some suggestions are provided in order to improve the work:
Abstract
The methodology used for creating the UHE index is not mentioned
Introduction
The case of Bejing is not described. Why was it considered as a success?
Figure 1 should be improved with some information, such as the UHE levels of the cities involved in the program
General comments:
clarify if the proposed framework for assessing the UHE is the one already developed by the goverment or it is an improved version of it.
from lines 46 to 65 there is a brief literature review that refers to the urban development topic, but if the Authors are able to improve this part with other studies that have already addressed the creation of similar assessment index (such as: Anelli, D., Tajani, F., & Ranieri, R. (2022). Urban resilience against natural disasters: Mapping the risk with an innovative indicators-based assessment approach. Journal of Cleaner Production, 371, 133496 or Bakar, A. H. A., & Cheen, K. S. (2013). A framework for assessing the sustainable urban development. Procedia-Social and Behavioral Sciences, 85, 484-492.) it should be better.
In Table 1 the datat source and the reference years are missing
The innovative contributions and the practical implications should be better highlighted and deeply discussed
Author Response
Point 1: Abstract
The methodology used for creating the UHE index is not mentioned
Response 1: Thanks for the reviewer’s very supportive comments.
The methodology used for creating the UHE index has been added to the Abstract.
(In lines 12-13) Then, an evaluation index system of UHE incorporating 8 dimensions is constructed by referring to the existing authoritative indicator systems globally.
Point 2: Introduction:
(1)The case of Bejing is not described. Why was it considered as a success?
Response 2: Thanks for the reviewer’s very supportive comments.
The case of Beijing have been added to the Introduction.
During the pilot UHE in 2018, Beijing comprehensively monitored and evaluated the effect of the overall urban construction planning, and scientifically and systematically explored the problems existing in the urban construction process. The policy requirements of Beijing Master City Plan (2016-2035) is designed in accordance with the problems. As the first UHE pilot city, Beijing has successfully achieved the target of UHE, and provides a template for subsequent UHE work.
The details of revision work can be found in Lines 52-57.
Point 3: Figure 1 should be improved with some information, such as the UHE levels of the cities involved in the program
Response 3: Thanks for the reviewer’s very supportive comments.
We are really sorry, the data of cities involved in the program is confidential and only accessible to central government of China. But more necessary city information such as cities’ name has been added in Figure 1.
The details of revision work can be found in Figure 1.
Point 4: clarify if the proposed framework for assessing the UHE is the one already developed by the government or it is an improved version of it.
Response 4: Thanks for the reviewer’s very supportive comments.
The proposed framework for accessing the UHE is an improved version of it. Firstly, the UHE indicator system proposed by the government changes every year because the government's focus of work changes every year. Therefore, the indicator system proposed in this study only partially refers to the indicator system proposed by the government and also refers to other authoritative indicator systems around the world. Furthermore, the evaluation method and weighting method are both completely proposed by this study, the government did not provide any evaluation methods.
Point 5: from lines 46 to 65 there is a brief literature review that refers to the urban development topic, but if the Authors are able to improve this part with other studies that have already addressed the creation of similar assessment index (such as: Anelli, D., Tajani, F., & Ranieri, R. (2022). Urban resilience against natural disasters: Mapping the risk with an innovative indicators-based assessment approach. Journal of Cleaner Production, 371, 133496 or Bakar, A. H. A., & Cheen, K. S. (2013). A framework for assessing the sustainable urban development. Procedia-Social and Behavioral Sciences, 85, 484-492.) it should be better.
Response 5: Thanks for the reviewer’s very supportive comments.
Thank you very much for the literature recommended, which helps to make this part more convincing. These valuable literature have been added to the revised manuscript.
The details of revision work can be found in Lines 63-69.
Point 6: In Table 1 the data source and the reference years are missing
Response 6: Thanks for the reviewer’s very supportive comments.
Table 1 just describes a general indicator system without presenting any data. Data collection was carried out in case study section. The reference data sources for each indicator in Table 1 have been added to the section 4.2 Data source.
This study collected the data for each indicator (as listed in Table 1) of Wuhan City from 2010 to 2019 from statistical yearbooks and government bulletins, including Hubei Statistical Yearbook [55], Wuhan Statistical Yearbook[56], China City Statistical Yearbook[57], and Wuhan Environmental Status Bulletin[58].
[55]Hubei provincial Bureau of Statistics. Hubei Statistical Yearbook[J]. http://tjj.hubei.gov.cn/tjsj/sjkscx/tjnj/qstjnj/index.shtml.(Accessed May 26, 2023).
[56]Wuhan Bureau of Statistics. Wuhan Statistical Yearbook[J]. http://tjj.wuhan.gov.cn/tjfw/tjnj/.(Accessed May 26, 2023).
[57] National Bureau of Statistics of China. China City Statistical Yearbook[J]. https://data.cnki.net/v3/Trade/yearbook/single/N2022040095?zcode=Z024.(Accessed May 26, 2023).
[58] Wuhan Ecological Environment Bureau. Wuhan Environmental Status Bulletin[R]. http://hbj.wuhan.gov.cn/fbjd_19/xxgkml/zwgk/hjjc/hjzkgb/.(Accessed May 26, 2023).
Point 7: The innovative contributions and the practical implications should be better highlighted and deeply discussed.
Response 7: Thanks for the reviewer’s very supportive comments.
We have rewritten the conclusion of this paper and added the description of contributions and innovations of the literature.
(In lines 448-461) The main contributions of this study include two aspects. Theoretically, this study refers to medical theory and the process of human physical examination to construct indicator system comprehensively. Then, a scientific mechanism which combines deviation maximization method and obstacle analysis method is proposed to diagnose urban problems. Practically, this study combines empirical analysis to demonstrate how the UHE method can be used to conduct a "physical examination" of cities. The results of the analysis show that the UHE framework can effectively diagnose urban diseases that exist in the process of urban development. Additionally, this mechanism can be used to investigate the evolution of urban diseases over a period of time and explain the impact of national macro-control policies on urban development. This helps local governments to grasp the current status of urban development, accurately identify the shortcomings of cities, and implement targeted measures to improve the quality of the urban living environment.
Reviewer 2 Report

Minor editing of English language required
Author Response
Point 1: There is a study have similar idea with this research, namely, “A novel methodology (WM-TCM) for urban health examination: A case study of Wuhan in China”. What’s the difference between your study and this study?
Response 1: Thank you for reviewer’s very supportive comments.
The revised manuscript have added the previous research 'A novel methodology (WM-TCM) for urban health examination: A case study of Wuhan in China' into the Introduction section. This paper is a supplement to the previous paper. The previous research just proposed a method to evaluate the performance of individual indicator by combining the subjective and objective methods. This paper focuses on constructing a comprehensive diagnostic model for urban diseases to evaluate the overall performance of cities.
Point 2: There are a number of methods to determine weights. Why this study chose Deviation maximization weighting method to determine the weight of each indicator in the Table 1.
Response 2: Thank you for reviewer’s very supportive comments.
There are various weighting methods such as deviation maximization method, analytical hierarchy process (AHP), entropy method, genetic algorithms (GA), and neural networks (NN). This study adopts deviation maximization method to determine the weighting of indicators. This approach has the advantage of amplifying indication variances and making it simpler to identify abnormal indicators. Therefore, the deviation maximization method has been widely used to determine weightings of indicator and multi-objective decisions.
The details of revision work can be found in section 3.2.2.
Point 3: In section 3.2.1, the standardization of index data seems unreasonable, as we usually use the formal for standardization of positive indicator. Then, what’s the difference between Intermediate type indicator and interval type indicator.
Response 3: Thank you for reviewer’s constructive comments.
This study considers four different types of indicators, namely, positive type, negative type, intermediate type, and interval type. For each type indicator, we first convert it into a positive one, and then use formula 4 for data standardization.
Point 4: This study introduce the concept of “urban health examination” based on the lifeform theory. However, this theory seems not play an important role in the research method section and case study section.
Response 4: Thank you for reviewer’s very supportive comments.
The main purpose of introducing “urban health examination” into this article is to assist the construction of a comprehensive indicator system for urban physical examinations, as well as the establishment of a process for conducting examinations, including evaluations of the overall city and individual indicators. This have been demonstrated in Conclusion Section.
Point 5: In conclusion, the contribution or innovation of this study is not very sharp or attractive. Please improve the contribution of this study.
Response 5: Thank you for reviewer’s very supportive comments.
The contributions have been rewritten as shown in Conclusion section.
The main contributions of this study are two aspects. Theoretically, this study refers to medical theory and the process of human physical examination to construct indicator system comprehensively. Then, a scientific mechanism which combines deviation maximization method and obstacle analysis method is proposed to diagnose urban problems. Practically, this study combines empirical analysis to demonstrate how the UHE method can be used to conduct a "physical examination" of cities. The results of the analysis show that the UHE framework can effectively diagnose urban diseases that exist in the process of urban development. Additionally, this mechanism can be used to investigate the evoluation of urban diseases over a period of time and explain the impact of national macro-control policies on urban development. This helps local governments to grasp the current status of urban development, accurately identify the shortcomings of cities, and implement targeted measures to improve the quality of the urban living environment.
Point 6: There are a few grammar errors in this article. Please further revise and improve the language.
Response 6:Thank you for reviewer’s constructive comments.
We have made corrections to the grammatical errors in the article to make the language more standard.
Reviewer 3 Report
Sustainability
A new perspective on the evaluation of urbanization sustainability: urban health examination
Hang Yan and Zhijiang Liu
The paper makes an interesting and, in my opinion, original contribution to the scientific methodology to assess the health status of a city. It is very timely in the field of studies on the relationship between health, sustainability and urbanisation. I do not find any particular originality in stating that this methodology stems from an interpretation of the city as the human body, a concept historically quite used in urban planning (think of urban metabolism and the city life cycle). It may be of impact to frame the proposed method in the steps of assessment and diagnosis for preventive purposes practised in medicine (Fig.2-3-4). In particular, the procedures set out in the paper can be compared with the medical approach to arrive at a system of indicators whose values on the 'body' of the city under study are compared with statistically derived benchmarks. However, this approach, even in medicine, does not now exempt the consideration of qualitative approaches based on listening to the patient (holistic medicine and narrative medicine), approaches which in the case of the city refer to other methods that appeal to urban sociology. The latter are not the subject of this study but it might be theoretically important to recall them in the introduction, if only by referring to an earlier recent paper co-authored by Hang Yan: 'A novel methodology (WM-TCM) for urban health examination: A case study of Wuhan in China', Ecological Indicators 136 (2022) 108602. It is unclear why this is not mentioned in the text of the introduction and bibliography. This paper delves into passages not dealt with in the previous paper, and from a methodological point of view it is useful in bridging gaps, moreover it provides tools that can also be used in other fields of application of sustainability assessment methods. I refer to the explication of standardisation and normalisation procedures for parameters and the obstacle model.
Paragraph 3.2.1 Standardisation of original index data explicates a method of standardisation and normalisation of indicators attributable to 4 types, and the related benchmarking, I believe this is an original contribution. It is indeed interesting because in this type of assessment, one often has to deal with indicators of a different nature and with significant discrepancies between the magnitudes of the various index values.
I must clearly state that I am not an expert to judge the correctness of the detailed calculation procedures and the application of the mathematical optimisation method, deviation maximisation method and obstacle model. I therefore limit myself to emphasising the interest of the application of these methodologies to assess the 'health' condition of a city and the obstacles to achieving 'good health', and I appreciate the clarity in the logical exposition of the procedure followed.
Paragraph 4. Case study of urban health examination is set out with great clarity and well-calibrated in its various parts.
Turning to more detailed observations.
In the introduction the references to the context of Chinese planning and programming initiatives on the subject, I think could be expanded. I am certainly less expert than the authors, but for example these references seem interesting to me:
Yang J, Siri JG, Remais JV, et al. The Tsinghua-Lancet Commission on Healthy Cities in China: unlocking the power of cities for a healthy China. Lancet 2018; published online April 17. http://dx.doi.org/10.1016/ S0140-6736(18)30486-0.
Healthy cities initiative in China: Progress, challenges, and the way forward. The Lancet Regional Health - Western Pacific 2022;27: 100539 Published online 15 July 2022 https://doi.org/10.1016/j. lanwpc.2022.100539
Again in the introduction I have the feeling that the authors only wanted to refer to specific research to support the thesis expressed in lines 57-58: "However, the majority of current studies evaluate urban development from a particular standpoint". Perhaps this is correct in relation to the research discussed in the paper, however some reference to more general studies in the text between lines 45-65 would seem appropriate to me. For example, I think it would be appropriate to have references not only to Chinese literature, and also to have a closer look at the bibliography dealing with the relationship between urban planning and sustainability and urban health.
I would like to make these suggestions, but the authors are certainly capable of considering how to improve this part on the State of the Art.
Routledge Handbook of Planning for Health and Well-being. Shaping a sustainable and healthy future. Edited By Hugh Barton, Susan Thompson, Sarah Burgess, Marcus Grant. 2015
Shaping Neighbourhoods For Local Health and Global Sustainability. By Hugh Barton, Marcus Grant, Richard Guise. Routledge 2021
Wang L., Shuwen Liao and Xiaojing Zhao (2016), 'Exploration of Approaches and Factors of Healthy City Planning', Urban Planning International, Vol. 4.
On line 79 about urban metabolism in addition to reference 26 I would put a classic text such as:
Newman, P. (1999), "Sustainability and cities: Extending the metabolism model", Landscape and
Urban Planning, Vol. 44, pp. 219-226.
In order to establish a clearer relationship between the words on lines 114 - 116 with Fig. 3 I would use the same terms e.g. instead of “identify”: “Assess” and “Diagnose”.
Reference 38 quoted on line 137 I think is incomplete. It was perhaps https://unhabitat.org/global-urban-indicators-database.
There is a number slippage in the references from line 185 to the following.
Yes in the caption of fig.4 it is better to cite the source if already published.
Author Response
Point 1: I do not find any particular originality in stating that this methodology stems from an interpretation of the city as the human body, a concept historically quite used in urban planning (think of urban metabolism and the city life cycle). It may be of impact to frame the proposed method in the steps of assessment and diagnosis for preventive purposes practised in medicine (Fig.2-3-4). In particular, the procedures set out in the paper can be compared with the medical approach to arrive at a system of indicators whose values on the 'body' of the city under study are compared with statistically derived benchmarks. However, this approach, even in medicine, does not now exempt the consideration of qualitative approaches based on listening to the patient (holistic medicine and narrative medicine), approaches which in the case of the city refer to other methods that appeal to urban sociology.
Response 1: Thanks for the reviewer’s very constructive comments.
We are very agree with your opinions. The main purpose of introducing medical diagnosis into this article is to assist the construction of a comprehensive indicator system for urban health examinations, as well as the establishment of a mechanism for evaluations of the overall city and individual indicators. The main innovation of this study lies in the application of medical examination methods to construction an indicator system for urban health examination and the diagnostic mechanism for urban problems. Additionally, it is even more important to use this mechanism to discover the urban problems and investigate the evolution of urban diseases over a period of time.
Point 2: The latter are not the subject of this study but it might be theoretically important to recall them in the introduction, if only by referring to an earlier recent paper co-authored by Hang Yan: 'A novel methodology (WM-TCM) for urban health examination: A case study of Wuhan in China', Ecological Indicators 136 (2022) 108602. It is unclear why this is not mentioned in the text of the introduction and bibliography. This paper delves into passages not dealt with in the previous paper, and from a methodological point of view it is useful in bridging gaps, moreover it provides tools that can also be used in other fields of application of sustainability assessment methods. I refer to the explication of standardisation and normalisation procedures for parameters and the obstacle model.
Response 2: Thanks for the reviewer’s very supportive comments.
The revised manuscript have added the previous research 'A novel methodology (WM-TCM) for urban health examination: A case study of Wuhan in China' into the Introduction section. This paper is a supplement to the previous paper. The previous research just proposed a method to evaluate the performance of individual indicator by combining the subjective and objective methods. This paper focuses on constructing a comprehensive diagnostic model for urban diseases to evaluate the overall performance of cities.
Point 3: Paragraph 3.2.1 Standardisation of original index data explicates a method of standardisation and normalisation of indicators attributable to 4 types, and the related benchmarking, I believe this is an original contribution. It is indeed interesting because in this type of assessment, one often has to deal with indicators of a different nature and with significant discrepancies between the magnitudes of the various index values.
I must clearly state that I am not an expert to judge the correctness of the detailed calculation procedures and the application of the mathematical optimisation method, deviation maximisation method and obstacle model. I therefore limit myself to emphasising the interest of the application of these methodologies to assess the 'health' condition of a city and the obstacles to achieving 'good health', and I appreciate the clarity in the logical exposition of the procedure followed.
Response 3: Thank you for helping us point out the innovation standardization process and we have added this innovation into the Conclusion section. Furthermore, thank you for affirming the logic of the article.
Point 4: In the introduction the references to the context of Chinese planning and programming initiatives on the subject, I think could be expanded. I am certainly less expert than the authors, but for example these references seem interesting to me:
Yang J, Siri JG, Remais JV, et al. The Tsinghua-Lancet Commission on Healthy Cities in China: unlocking the power of cities for a healthy China. Lancet 2018; published online April 17. http://dx.doi.org/10.1016/ S0140-6736(18)30486-0.
Healthy cities initiative in China: Progress, challenges, and the way forward. The Lancet Regional Health - Western Pacific 2022;27: 100539 Published online 15 July 2022 https://doi.org/10.1016/j. lanwpc.2022.100539
Response 4: Thanks for the reviewer’s very supportive comments.
The research on healthy cities in the literature is very useful, which greatly expands the research background of this paper. We have updated the literature on this research.
The details of revision work can be found in Lines 41-49.
Point 5: Again in the introduction I have the feeling that the authors only wanted to refer to specific research to support the thesis expressed in lines 57-58: "However, the majority of current studies evaluate urban development from a particular standpoint". Perhaps this is correct in relation to the research discussed in the paper, however some reference to more general studies in the text between lines 45-65 would seem appropriate to me.
For example, I think it would be appropriate to have references not only to Chinese literature, and also to have a closer look at the bibliography dealing with the relationship between urban planning and sustainability and urban health.
I would like to make these suggestions, but the authors are certainly capable of considering how to improve this part on the State of the Art.
Routledge Handbook of Planning for Health and Well-being. Shaping a sustainable and healthy future. Edited By Hugh Barton, Susan Thompson, Sarah Burgess, Marcus Grant. 2015
Shaping Neighbourhoods For Local Health and Global Sustainability. By Hugh Barton, Marcus Grant, Richard Guise. Routledge 2021
Wang L., Shuwen Liao and Xiaojing Zhao (2016), 'Exploration of Approaches and Factors of Healthy City Planning', Urban Planning International, Vol. 4.
Response 5: Thank you for recommending these references, they are very helpful to us. Based on these literature, we have rewritten the literature review section and removed the inappropriate statement “However, the majority of current studies evaluate urban development from a particular standpoint”.
The details of revision work can be found in Lines 63-73
Point 6: On line 79 about urban metabolism in addition to reference 26 I would put a classic text such as:
Newman, P. (1999), "Sustainability and cities: Extending the metabolism model", Landscape and Urban Planning, Vol. 44, pp. 219-226.
Response 6: Thank you very much for your recommendation. We have learned a lot from this literature. It is very helpful and have been added in Section 2.
The details of revision work can be found in Lines 105-107.
Point 7: In order to establish a clearer relationship between the words on lines 114 - 116 with Fig. 3 I would use the same terms e.g. instead of “identify”: “Assess” and “Diagnose”.
.
Response 7: Thanks for the reviewer’s very supportive comments.
We all used the term of “Diagnose” in the literature to be consistent with Figure 3.
The details of revision work can be found in Lines 142-145.
Point 8: Reference 38 quoted on line 137 I think is incomplete. It was perhaps https://unhabitat.org/global-urban-indicators-database.
Response 8: Thanks for the reviewer’s very supportive comments.
We checked the website link of the data and supplemented the link information of the data.
[47]UN-Habitat. (2014). best-practices database. https://unhabitat.org/global-urban-indicators-database (Accessed May 14, 2022).
Point 9: There is a number slippage in the references from line 185 to the following.
Response 9: Thanks for the reviewer’s very supportive comments.
The reference in the wrong location has been modified.
Point 10: Yes in the caption of fig.4 it is better to cite the source if already published.
Response 10: Thanks for the reviewer’s very supportive comments.
The source of the data has been indicated in Figure 4.
Round 2
Reviewer 1 Report
The efforts made by the Authors are apprecciated